# Risk factors associated with self-medication among the indigenous communities of Chittagong Hill Tracts, Bangladesh

**Ayan Saha**[1]*, **Kay Kay Shain Marma**[2], **Afrah Rashid**[3☯], **Nowshin Tarannum**[4☯], **Srabanty Das**[5], **Tonmoy Chowdhury**[6], **Nusrat Afrin**[2], **Prashanta Chakraborty**[7], **Md. Emran**[8], **H. M. Hamidullah Mehedi**[9], **Mohammad Imdad Hussain**[10], **Ashim Barua**[11], **Sabuj Kanti Mistry**[12,13,14]

**1** Department of Genetic Engineering and Biotechnology, East West University, Dhaka, Bangladesh, **2** Department of Pharmacy, University of Chittagong, Chattogram, Bangladesh, **3** Department of Public Health and Bioinformatics, Asian University for Women, Chattogram, Bangladesh, **4** Department of Microbiology, University of Chittagong, Chattogram, Bangladesh, **5** Department of Biochemistry and Molecular Biology, University of Chittagong, Chattogram, Bangladesh, **6** Rangamati Medical College, Rangamati, Bangladesh, **7** Department of Genetic Engineering and Biotechnology, University of Chittagong, Chattogram, Bangladesh, **8** Department of Biochemistry & Biotechnology, University of Science & Technology Chittagong, Chattogram, Bangladesh, **9** Department of Medicine, 250 Beded General Hospital, Chattogram, Bangladesh, **10** Upazila Health Complex, Rowangchari, Bandarban, Bangladesh, **11** Upazila Health Complex, Alikadam, Bandarban, Bangladesh, **12** ARCED Foundation, Mirpur, Dhaka, Bangladesh, **13** Centre for Primary Health Care and Equity, University of New South Wales, Sydney, Australia, **14** Department of Public Health, Daffodil International University, Dhaka, Bangladesh

☯ These authors contributed equally to this work.
* ayan.saha.bd@gmail.com, ayan.saha@ewubd.edu

**Data Availability Statement:** All relevant data are within the paper and its Supporting information files.

**Funding:** The author(s) received no specific funding for this work.

## Abstract

### Background

In developing countries like Bangladesh, self-medication has become a predicament associated with health risks and clinical complications. To date, no studies have been conducted on the practice of self-medication among the indigenous population living in Chittagong Hill Tract (CHT).

### Objectives

This study was aimed to determine the prevalence of self-medication and analyzing the factors associated with it among the indigenous population in CHT.

### Methods

This cross-sectional study was conducted from late October to early December 2020; among different indigenous group populations residing in the three districts of CHT aged 18 or more. A pre-tested and semi-structured questionnaire was developed to collect data on socio-demographic characteristics, health status, frequency of self-medication, reasons for self-medication in last one year, as well as other variables. Multivariate logistic regression was performed to assess associated factors with self-medication.

**Competing interests:** The authors have declared that no competing interests exist.

## Results

A total of 1350 people from different indigenous populations were interviewed, among whom 49.9% practiced self-medication. The rate of self-prescribed antibiotics usage (80.9%) was significantly higher compared to other drugs. Self-prescribed medications were mostly used for diarrhea and food poisoning (60.6%), cough, cold and fever (51.4%), and headache (51.4%). A common source of self-prescribed medicines was community or retail pharmacy and the most reported reason for self-prescribed medication was the long-distance of healthcare facilities from home.

## Conclusion

The prevalence of self-medication is substantially high among indigenous people and the effect is alarming. Particular concern is the misuse of antibiotics and analgesic drugs. Increasing awareness among the population of the negative effect of self-medication and implementation of proper policies and actions are urgently needed to prevent self-medication among indigenous population in Bangladesh.

## Introduction

According to the World Health Organization (WHO), when an individual consumes medicines based on their self-diagnosis of a disease, without consulting a medical practitioner or taking any clinical assays to justify their assumptions it is termed as self-medication [1]. Self-medication does not only refer to consuming medicines based on acute symptoms, rather, it also involves repetitive self-administration of medicines for chronic diseases [2]. Evidence suggest that people who adhere to self-medication themselves have also been found to advise their family members, relatives, and friends to do the same [3]. Self-medication is a major healthcare concern as it may result in various detrimental effects such as misdiagnosis of the illness, antimicrobial resistance, harmful drug interactions, or even delay in the diagnosis of a serious disease [4].

Self-medication has now become a global practice, where people tend to buy over-the-counter drugs merely based on symptom mapping. The active and passive effects of self-medication on health management have enlisted it as a global public health concern [5]. Studies conducted in Europe reported at least 21% of the population took self-medication [6]. The prevalence rate for self-medication was found to be 38.8% and 75.7% in Asia [7] and Africa [8] respectively. In Bangladesh, the practice of self-medication has reached to a point where it is quite normal for people to consume and reuse prescription drugs given any onset of health aberration, like fever, nausea, fatigue etc. In earlier studies, 16.0%–81.4% prevalence of self-medication [4, 9, 10] and 26.7% prevalence of self-medication with antibiotics [11] were reported among urban populations in Bangladesh. Similar observations were reported in neighboring countries and the prevalence rates for self-medication were found to be 12.0%–78.6% in India [12–14], 38.2%–82.0% in Nepal [15–17], 35.3%–78.0% in Sri Lanka [18–20] and 83.0% in Iran [21]. Studies have found the most common reasons for self-prescribed medication to be the prior experience of the illness, inadequate information about the illness, financial problems for visiting a physician, insufficient time, and easy access to medications [22, 23].

According to the United Nations, indigenous people are recognized as the most "vulnerable, disadvantaged and marginalized people" [24]. There are 3 million indigenous peoples lives in Bangladesh consisting 54 different indigenous community speaking 35 distinctive language, which make up approximately 2% of the total population of Bangladesh [25]. They are mostly located in the remote southern part of the country, known as the CHT (Chittagong Hill Tracts), where the tribes called Chakma, Marma, and Tripura are found to be of the vast majority [26]. As majority of the indigenous community live in remote locations, quite at a distance from urban areas, they are often deprived of receiving adequate healthcare services [25]. Studies have also reported that indigenous populations suffer more from disease burden of communicable diseases and health inequalities than mainstream populations [26].

Research has documented a high prevalence of 54.9%–92.1% self-medication practice among the populations living in India's hilly areas [27, 28]. Although several studies have been conducted previously on the tendency and prevalence of self-medication among different Bangladeshi populations [4, 9–11, 23, 29–31], the attitude of the indigenous population of Bangladesh towards this severely concerning public health issue has not been explored.

Therefore, in this study, we investigated the prevalence of self-medication among a range of indigenous communities living in Chittagong Hill Tracts. We further analyzed demographic and economic determinants, which might have a role in driving communities towards self-medication.

## Materials and methods

### Study design, setting, and participants

This cross-sectional study was carried out amongst the indigenous population of Bangladesh, living in the CHT, located in the southern part of the country. CHT comprises three hilly district areas: Rangamati, Bandarban, and Khagrachari. The sample size of this study was calculated using the formula, $n = z^2 p(1 − p)/d^2$ [32], considering a significance level of 0.05, a confidence interval of 95%, a 75% prevalence rate collected from an earlier study in Bangladesh [29], 5% margin of error, a 20% non-response rate and finally multiplied by a design effect = 2 due to cluster sampling technique. Using the aforementioned formula, the minimum required number of participants was calculated to be 692. However, to ensure maximum variations among the study participants, we aimed to have as many participants as possible and finally ended up with 1350. The clusters were based on the subunits of each of the three districts. A proportionate simple random sampling technique was used to select an equivalent number of participants from each subunit of the three districts. The inclusion criteria included indigenous participants and those of age $\geq$ 18 years.

### Measures

The outcome variables for this study were the self-medication practices among CHT's indigenous population. Self-medication was defined as the use of over-the-counter medications as well as the use of medications previously prescribed by doctor for a disease and taking the same medications for the current episode of that disease without consulting a doctor. Participants were asked if they had taken medicine in the preceding year without the advice of a doctor or a healthcare professional.

### Explanatory variables

Explanatory variables included Administrative District (Rangamati, Bandarban, and Khagrachari), age (dichotomized as 18–35, 36–50,51–65 and 65 above), sex (male and female),

ethnicity (Chakma, Marma, Tripura, Bawm, Tanchangya, Chak, Rakhine, Saotal and Mro), marital status (married, never married, widow/widower, separated/divorced), educational level (illiterate, primary, secondary, higher secondary and graduate), occupation (agricultural work, service, housewife, student, business, day labor, handloom and unemployed) family income in US dollar (USD) (<236 USD, 236–591 USD, >591 USD). Few other variables, such as body mass index (BMI) kg/m$^2$ ((<18.5 (Underweight), 18.5–23.5 (Normal), >23.5 (Overweight)), common disease prevalence (cough, cold and fever, headache, joint pain, diarrhea and food poisoning, dental carries and toothache, irritable bowel syndrome, typhoid, malaria, jaundice, roundworm/tapeworm, sinusitis, asthma, other respiratory diseases, acne, skin allergy and none), common comorbidities (eye problem, anxiety disorder, skin infection, hypertension, hypotension, diabetes, respiratory diseases, liver disease, cancer, neurological disorders, heart disease, kidney disease, thyroiditis and none) and types of medication (antipyretics, analgesic, antibiotics, antacids and anti-ulcerants, antidiarrheal, antitussive, anti-allergic, vitamins, antiemetic, sedatives, contraceptives, insulin, beta blockers, steroids), were taken into consideration in terms of disease prevalence associated self-medication.

## Data collection

A validated structured questionnaire consisting of 33 questions on self-medication behavior was used to collect information between late October and early December 2020 through face-to-face interviews. The questionnaire consisted of two sections, where section 1 assessed the socio-demographic variables of the participants and their epidemiological characteristics that included lifestyle, comorbidities, frequently occurring diseases, and whether or how often they took antibiotics or any other drugs.

Section 2 took into account the participants' frequency of medical consultancy. They were also asked about the minimal distance between their residence and the closest medical center as well as the closest pharmacy and finally their average monthly family income. The questionnaire was reviewed and evaluated by professionals including, an epidemiologist, a public health expert, a pharmacist, medicine specialists, and doctors.

The questionnaire was prepared in English and then back-translated into Bengali (S1 and S2 Files). We recruited 7 research assistants locally who understand the dialect of the specific indigenous tribes and extensively trained them before the data collection. All the data were first manually recorded in paper and then transferred to Google forms software to be exported and stored in Microsoft Excel 2013.

## Statistical analysis

We assessed the distribution of the socio-demographic variables through descriptive analysis. The results were graphically represented using GraphPad prism (version 9.0). With a 5% threshold of significance, Chi-square tests were employed to compare the differences in the prevalence of self-medication by the variables. A binary logistic regression model was employed to measure the association between self-medication and explanatory variables. The variables with a p value of less than 0.25 in the unadjusted analysis was included in the final multiple regression model [33]. Both unadjusted and adjusted odds ratios (OR) are reported with a 95% confidence interval (95% CI). A p-value of 0.05 was considered significant in the final model. All analyses were performed using the statistical software package SPSS (version 25.0).

### Ethical consideration

The study was approved (Ref: 1728) by the Institutional review board of 250 bedded General Hospital, Chattogram, Bangladesh. All the participants participated voluntarily with verbal consent as some of the participants were unable to read and write, and no payment was offered.

## Results

### Socio-demographic characteristics

A total of 1350 individuals from three hill districts–Rangamati (39.9%), Khagrachhari (33.5%), and Bandarban (26.7%) participated in this study which included major indigenous (Chakma, Marma, Tripura, Tanchangya) and minor indigenous (Bawn, Chak, Saotal) populations living in CHT. The majority of the study participants, 834 (61.8%) were aged between 18–35 years and the male (50.9%), female (49.1%) ratio was almost equal. Over three-quarters of the participants had received primary education or higher and more than half (55.8%) had limited monthly income (<236 USD). Various types of socio-demographic characteristics are presented in Table 1.

Among the total 1350 participants, half of them (49.9%) including 57.6% of male and 41.9% of female reported to practice self-medication. A high prevalence of self-medication was observed among the 18–35 (54.7%) years age cohort. The frequency of self-medication practice has been found more prevalent among students (61.1%) and people who are graduate (60.1%) comparatively than others (Table 1).

### Prevalence of common diseases, comorbidities, and self-medication practice

Cough, cold, and fever were found to be highly prevalent common diseases among the target population (S1A Fig), and results from three of the districts showed a similar rate of prevalence of diseases (S1B Fig). Prevalence of joint pain was observed more in older people (> 65 years) and in populations with high BMI (S1C and S1D Fig).

In terms of comorbidities, Eye problem was commonly predominant in each three of the districts (S2A Fig). People from Khagrachari district reported more hypertensive and people from Bandarban had more percentage of hypotensive patients than the other two districts (S2B Fig). Hypertension was found prevalent among people from the 51–65 age range and those who had a BMI of more than 23.5. On the other hand, hypotension was prevalent among people about 36–50 year and BMI less than 18.5 (S2C and S2D Fig).

In the last 5 years, the most commonly reported health complications were cough, cold, and fever (66.7%), following headache (34.1%), joint pain (17.4%), diarrhea and food poisoning (14.3%), dental caries, and toothache (13.8%), irritable bowel syndrome (13.2%) and others (31.6%) (Table 2). About 60.6% of people who suffered from diarrhea had taken self-medication and around half of the population suffering from other diseases had practiced self-medication. The most common comorbidity found among the population was eye problem (16.7%), anxiety disorder (12.9%), and skin infection (10.7%) respectively. More than 60% of cases of self-medication were reported among the study population with different comorbidities (Table 2). Antipyretics (52.6%) and analgesics (47.4%) were the most used medicines and 53.6% and 48% of them were taken by self-prescription respectively. There was an alarming frequency of self-medication when taking antibiotics (80.9%) (Table 2). It is observed that people associated with agricultural work were using more antipyretics, analgesics, and antibiotics than service holders (S1 Table).

**Table 1. Prevalence of self-medication and socio-demographic characteristics of study participants (n = 1350).**

| Variable | Group | Frequency n = 1350 (%) | Self-medication | | $\chi^2$ p-value |
|---|---|---|---|---|---|
| | | | Yes n = 674 (%) | No n = 676 (%) | |
| Age (34.64±14.84) | 18–35 | 834 (61.8%) | 456 (54.7%) | 378 (45.3%) | 0.001 |
| | 36–50 | 310 (23.0%) | 126 (40.6%) | 184 (59.4%) | |
| | 51–65 | 151 (11.2%) | 67 (44.4%) | 84 (55.6%) | |
| | >65 | 55 (4.1%) | 25 (45.5%) | 30 (54.5%) | |
| Gender | Male | 687 (50.9%) | 396 (57.6%) | 291 (42.4%) | 0.001 |
| | Female | 663 (49.1%) | 278 (41.9%) | 385 (58.1%) | |
| Home District | Rangamati | 538 (39.9%) | 285 (53.0%) | 253 (47.0%) | 0.001 |
| | Khagrachhari | 452 (33.5%) | 282 (62.4%) | 170 (37.6%) | |
| | Bandarban | 360 (26.7%) | 107 (29.7%) | 253 (70.3%) | |
| Ethnicity | Chakma | 551 (40.8%) | 319 (57.9%) | 232 (42.1%) | 0.001 |
| | Marma | 481 (35.6%) | 237 (49.3%) | 244 (50.7%) | |
| | Tripura | 128 (9.5%) | 47 (36.7%) | 81 (63.3%) | |
| | Bawm | 75 (5.6%) | 18 (24.0%) | 57 (76.0%) | |
| | Tanchangya | 73 (5.4%) | 34 (46.6%) | 39 (53.4%) | |
| | Others* | 42 (3.1%) | 19 (45.2%) | 23 (54.8%) | |
| Marital status | Married | 722 (53.5%) | 305 (42.2%) | 417 (57.8%) | 0.001 |
| | Never married | 577 (42.7%) | 347 (60.1%) | 230 (39.9%) | |
| | Widow/widower | 45 (3.3%) | 18 (40.0%) | 27 (60.0%) | |
| | Separated/Divorced | 6 (0.4%) | 4 (66.7%) | 2 (33.3%) | |
| Educational level | Illiterate | 200 (14.8%) | 74 (37.0%) | 126 (63.0%) | 0.001 |
| | Primary | 198 (14.7%) | 77 (38.9%) | 121 (61.1%) | |
| | Secondary | 215 (15.9%) | 106 (49.3%) | 109 (50.7%) | |
| | Higher secondary | 354 (26.2%) | 187 (52.8%) | 167 (47.2%) | |
| | Graduate | 383 (28.4%) | 230 (60.1%) | 153 (39.9%) | |
| Occupation | Agricultural work | 254 (18.8%) | 98 (38.6%) | 156 (61.4%) | 0.001 |
| | Service | 185 (13.7%) | 88 (47.6%) | 97 (52.4%) | |
| | Housewife | 214 (15.9%) | 72 (33.6%) | 142 (66.4%) | |
| | Student | 473 (35.0%) | 289 (61.1%) | 184 (38.9%) | |
| | Others** | 224 (16.6%) | 127 (56.7%) | 97 (43.3%) | |
| Income (USD) | <236 USD | 753 (55.8%) | 372 (49.4%) | 381 (50.6%) | 0.580 |
| | 236–591 USD | 521 (38.6%) | 261 (50.1%) | 260 (49.9%) | |
| | >591 USD | 76 (5.6%) | 41 (53.9%) | 35 (46.1%) | |

*Others in ethnicity includes: Chak, Rakhine, Saotal and Mro.

**Others in occupation includes: Business, Day labor, Handloom and Unemployed.

Among the people with a tendency of self-medication, 60.2% reported practicing self-medication at least once a year and 32.9% of them had taken medication 2–5 times/ year without prescription (Fig 1A). Their common source of self-medication was local community pharmacy (48%). Some individuals took medicines that were suggested by friends and family members while others consumed particular drugs frequently based on prescriptions from earlier treatment phases (Fig 1B). Antipyretics (56.3%), analgesics (45.5%) and antibiotics (34.5%) were the most used medication among self-medicated people (Fig 1C). A clear differentiation has been identified between the minimal distance of local community pharmacy and hospital from home suggesting that the long-distance of the hospitals may provoke indigenous people to take self-prescribed medication from the pharmacies that are closer to them (Fig 1D).

**Table 2. Disease prevalence and self-medication practice among indigenous people.**

| Variable Group | Frequency; n = 1350 (%) | Self-medication | | p-value |
|---|---|---|---|---|
| | | Yes; n = 674 (%) | No; n = 676 (%) | |
| **Body mass index (BMI) kg/m$^2$** | | | | |
| <18.5 (Underweight) | 249 (18.4%) | 111 (44.6%) | 138 (55.4%) | **0.059** |
| 18.5–23.5 (Normal) | 742 (55.5%) | 368 (49.6%) | 374 (50.4%) | |
| >23.5 (Overweight) | 359 (26.6%) | 195 (54.3%) | 164 (45.7%) | |
| **Common disease prevalence** | | | | |
| Cough, cold and fever | 901 (66.7%) | 463 (51.4%) | 438 (48.6%) | **0.058** |
| Headache | 461 (34.1%) | 237 (51.4%) | 224 (48.6%) | |
| Joint pain | 235 (17.4%) | 117 (49.8%) | 118 (50.2%) | |
| Diarrhea and food poisoning | 193 (14.3%) | 117 (60.6%) | 76 (39.4%) | |
| Dental carries and toothache | 186 (13.8%) | 95 (51.1%) | 91 (48.9%) | |
| Irritable bowel syndrome | 178 (13.2%) | 89 (50%) | 89 (50%) | |
| Others* | 426 (31.6%) | 222 (52.1%) | 204 (47.9%) | |
| None | 177 (13.1%) | 84 (47.5%) | 93 (52.5%) | |
| **Common comorbidities** | | | | |
| Eye problem | 226 (16.7%) | 136 (60.2%) | 90 (39.8%) | **0.001** |
| Anxiety disorder | 174 (12.9%) | 114 (65.5%) | 60 (34.5%) | |
| Skin infection | 145 (10.7%) | 91 (62.8%) | 54 (37.2%) | |
| Hypertension | 118 (8.7%) | 75 (63.6%) | 43 (36.4%) | |
| Hypotension | 100 (7.4%) | 48 (48%) | 52 (52%) | |
| Diabetes | 40 (3.0%) | 24 (60%) | 16 (40%) | |
| Respiratory diseases | 47 (3.5%) | 30 (63.8%) | 17 (36.2%) | |
| Others** | 92 (6.8%) | 52 (56.5%) | 40 (43.5%) | |
| None | 709 (52.5%) | 295 (41.6%) | 414 (58.4%) | |
| **Types of medication** | | | | |
| Antipyretics | 710 (52.6%) | 380 (53.6%) | 330 (46.4%) | **0.001** |
| Analgesic | 640 (47.4%) | 307 (48.0%) | 333 (52.0%) | |
| Antibiotics | 288 (21.3%) | 233 (80.9%) | 55 (19.1%) | |
| Antacids and Anti-ulcerants | 290 (21.5%) | 158 (54.5%) | 132 (45.5%) | |
| Antidiarrheal | 251 (18.6%) | 117 (46.6%) | 134 (53.4%) | |
| Antitussive | 181 (13.4%) | 108 (59.7%) | 73 (40.3%) | |
| Anti-allergic | 124 (9.2%) | 85 (68.5%) | 39 (31.5%) | |
| Vitamins | 116 (8.6%) | 65 (56.0%) | 51 (44.0%) | |
| Antiemetic | 61 (4.5%) | 29 (47.5%) | 32 (52.5%) | |
| Sedatives | 31 (2.3%) | 14 (45.2%) | 17 (54.8%) | |
| Others*** | 46 (3.4%) | 30 (65.2%) | 16 (34.8%) | |
| Not answered | 203 (15.0%) | 71 (35.0%) | 132 (65.0%) | |

*Others in common disease includes: Typhoid, Malaria, Jaundice, Roundworm/Tapeworm, Sinusitis, Asthma, Other respiratory diseases, Acne, Skin allergy

**Others in comorbidities includes: Liver disease, Cancer, Neurological disorders, Heart disease, Kidney disease, Thyroiditis

***Others in medicine type includes: Contraceptives, Insulin, Beta blockers, Steroids

## Factors associated with self-medication

Table 3 shows the factors associated with self-medication. We have considered all the socio-demographic characteristics of the participants presented in Table 1 in regression analysis. At bivariate analysis, Chakma (Crude Odd Ratio (cOR): 16.05, 95% CI: 2.19–117.68,

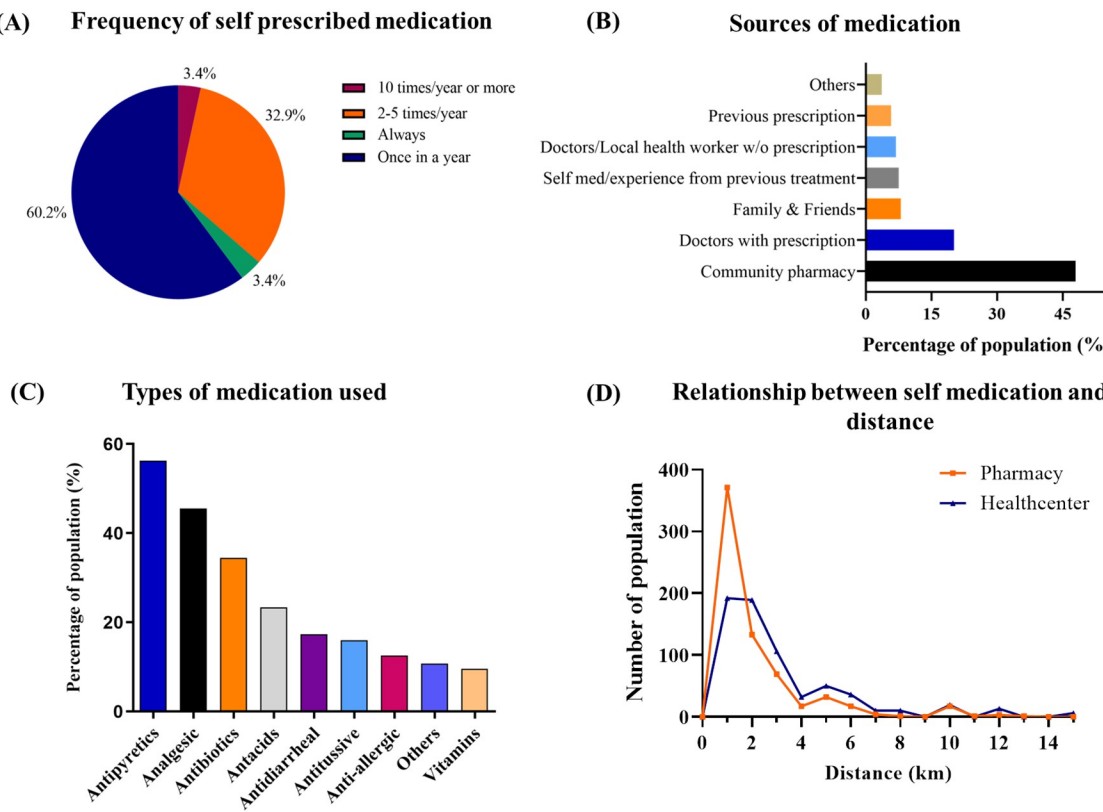

**Fig 1. Characteristics of self-medication practice among self-medicated people.** (A) representing how often respondents self-medicated; (B) representing the different sources of self-medication where X-axis indicates the percentage of population and Y-axis indicates sources of medication; (C) representing the most common self-prescribed medications where the X-axis represents types of medication and Y-axis represents the percentage of the population; lastly (D) representing the relation between self-medication and distance from home to the closest pharmacy and healthcare center; here Y-axis represents the number of the population and X-axis represents the distance (km) of pharmacy and hospital.

P = 0.006) and Marma (cOR: 10.36, 95% CI: 1.41–76.24, P = 0.022) participants were more likely to self-medicate than other minor indigenous groups. Based on literacy rates, participants who were illiterate (cOR: 0.24, 95% CI: 0.14–0.40, P = 0.001), educated up to the primary (cOR: 0.25, 95% CI: 0.15–0.42, P = 0.001), secondary (cOR: 0.39, 95% CI: 0.25–0.61, P = 0.001) and higher secondary levels (cOR: 0.70, 95% CI: 0.50–0.97, P = 0.032) were less likely to self-medicate than graduate participants (Table 3). Based on the participants' profession, agricultural workers (cOR: 0.38, 95% CI: 0.23–0.65, P = 0.001) and housewives (cOR: 0.42, 95% CI: 0.24–0.72, P = 0.002) were less likely to be inclined towards self-medication than most other occupation groups. All the variables had a p value less than 0.25 in the bivariate analysis and thus all of them were included in the multivariable model. Hence, the model is adjusted for all the variables in Table 3. In multivariable logistic regression, female participants (Adjusted Odd Ration (aOR) = 0.54, 95% CI:0.40 to 0.76, P = 0.001) were less likely to take medications without prescriptions than their male counterparts. Participants from Rangamati (aOR = 4.83, 95% CI: 2.66–8.80, P = 0.001) and Khagrachari (aOR = 4.74, 95% CI: 2.53–8.88, P = 0.001), on the other hand, were more likely to practice self-medication than participants from Bandarban.

**Table 3. Factors associated with self-medication.**

| Variables | Odds Ratio | | | |
|---|---|---|---|---|
| | Crude (95% CI) | p-value | Adjusted (95% CI) | p-value |
| **Age** | | | | |
| 18–35 | 1.92 (0.89–4.13) | **0.095** | 0.60 (0.22–1.67) | **0.331** |
| 35–50 | 0.58 (0.25–1.36) | **0.211** | 0.45 (0.17–1.20) | **0.107** |
| 61–65 | 1.52 (0.65–3.54) | **0.334** | 1.08 (0.42–2.80) | **0.868** |
| >65 (ref) | 1.00 | | 1.00 | |
| **Gender** | | | | |
| Female | 0.53 (0.41–0.69) | **0.001** | 0.54 (0.40–0.76) | **0.001** |
| Male (ref) | 1.00 | | 1.00 | |
| **District** | | | | |
| Rangamati | 8.16 (4.77–13.94) | **0.001** | 4.83 (2.66–8.80) | **0.001** |
| Khagrachari | 6.83 (3.96–11.79) | **0.001** | 4.74 (2.53–8.88) | **0.001** |
| Bandarban (ref) | 1.00 | | 1.00 | |
| **Ethnicity** | | | | |
| Chakma | 16.05 (2.19–117.68) | **0.006** | 3.81 (0.49–29.47) | **0.201** |
| Marma | 10.36 (1.41–76.24) | **0.022** | 4.42 (0.58–33.89) | **0.153** |
| Tripura | 4.64 (0.59–36.55) | **0.145** | 2.07 (0.25–17.20) | **0.501** |
| Bawm | 0.000 | **0.997** | 0.000 | **0.997** |
| Tanchangya | 4.35 (0.52–36.64) | **0.176** | 2.42(0.27–21.69) | **0.429** |
| Others* (ref) | 1.00 | | 1.00 | |
| **Marital Status** | | | | |
| Married | 0.26 (0.05–1.42) | **0.933** | 0.85 (0.13–5.62) | **0.863** |
| Never married | 0.93 (0.17–5.12) | **0.119** | 0.22 (0.04–1.43) | **0.114** |
| Widow | 0.31 (0.05–2.06) | **0.225** | 0.25 (0.03–1.97) | **0.187** |
| Divorced (ref) | 1.00 | | 1.00 | |
| **Educational Status** | | | | |
| Illiterate | 0.24 (0.14–0.40) | **0.001** | 0.47 (0.22–1.02) | **0.057** |
| Primary | 0.25 (0.15–0.42) | **0.001** | 0.51 (0.25–1.01) | **0.053** |
| Secondary | 0.39 (0.25–0.61) | **0.001** | 0.77 (0.44–1.33) | **0.342** |
| Higher secondary | 0.70 (0.50–0.97) | **0.032** | 0.80 (0.56–1.14) | **0.220** |
| Graduate (ref) | 1.00 | | 1.00 | |
| **Occupation** | | | | |
| Agricultural work | 0.38 (0.23–0.65) | **0.001** | 1.04 (0.55–1.98) | **0.902** |
| Service | 0.80 (0.49–1.30) | **0.365** | 0.98 (0.55–1.74) | **0.931** |
| Housewife | 0.42 (0.24–0.72) | **0.002** | 1.30 (0.66–2.53) | **0.459** |
| Student | 1.64 (1.13–2.38) | **0.010** | 0.64 (0.38–1.09) | **0.098** |
| Others** (ref) | 1.00 | | 1.00 | |
| **Income** | | | | |
| <236 USD | 1.24 (0.65–2.36) | **0.513** | 2.41 (1.21–4.80) | **0.012** |
| 236–591 USD | 1.60 (0.83–3.02) | **0.168** | 3.11 (1.57–6.20) | **0.001** |
| >591 USD (ref) | 1.00 | | 1.00 | |

*Other in ethnicity includes: Chak, Rakhine, Saotal and Mro

**Others in occupation includes: Business, Day labor, Handloom and Unemployed

## Discussion

This paper highlighted about the self-medication practices among the indigenous people of Bangladesh which can pose many inherent risks to them. This is especially true for some patient populations such as children or adults who are more likely to be exposed to drug-related clinical complications [34]. This study shows that half of the study population (49.9%) had taken medication without a prescription at least once or more in a year. This percentage is higher in comparison with the result of a study conducted on participants of Chittagong city in which 41% of the study population confirmed taking self-medication [23]. This clearly shows the high prevalence of self-medication in hill tracts than the urban areas of the same geographic location. This prevalence is also very high compared to that of an urban city of punducherry, India which reported a self-medication prevalence rate of about 11.9%. The prevalence was also higher compared to the prevalence of Brazil (18.3%), Srilanka (12.2%), Portugal (26.2%), Saudi Arabia (35.4%), and Iran (35.4%) [19, 35–38]. However, In tribal districts of the south region, India (54.9%) and in Northwest Ethiopia (50.2%) a similar prevalence rate of self-medication was observed [13, 28, 39]. Moreover, the prevalence of self-medication that we found among the indigenous groups of CHT (49.9%) are significantly lower than results observed in western Nepal (59%), south India (92%), Srilanka (60.8%), Serbia (79.9%), Italy (69.2%), Turkey (63.4%), Egypt (62.9%) and Pakistan (76%) [17, 18, 40–45]. Different target populations, socio-economical characteristics, knowledge about specific diseases, cultural variations could be associated with the difference among these results. The findings of the present study impose an urgency of careful monitoring and regulation of the drug consumption, drug delivery, drug dispensing in developing countries like Bangladesh to prevent self-medication.

Male participants reported about 1.37 times more likely to self-medicate in comparison with women and surprisingly highly educated people i.e., graduates (60.1%) were practicing more self-medication. These findings deviate from the results found from a similar study in India where, in some hilly areas where the prevalence of self-medication was 87.4% among illiterate people [46]. Studies on self-medication practice conducted among the indigenous population in North Maharashtra (92.1%), Vietnam (83.3%), Meghalaya (55%) confer about the knowledge and attitude towards health-seeking behaviors [27, 47, 48]. It has been noticed that a high percentage of educated people preferred self-medication and they tend to do it in spite of knowing the side effects. This could be because some of the educated people consider themselves knowledgeable about medications and hence do not hesitate to self-medicate [49]. A recent study among undergraduates reported that students had a positive attitude about self-medication though they were aware of adverse effects [50]. Among students, the main causes of self-medication were found to be underestimation of the need for professional advice for minor illnesses, hospital-related anxiety, past success with self-medication which were readily available over-the-counter medicines, and friends/family and pharmacists' (OTC) recommendations [51, 52].

In this study, the local community pharmacy was found to be the most common primary source of self-medication. The distance of the healthcare facilities can play a significant role in motivating people to consume drugs without medical supervision from the nearest pharmacies [17]. The present study also revealed that the nearest community pharmacies were available to most of the participants within 1.5 km from their residence and healthcare facilities were about 2–4 km away. An earlier study showed that long-distance hospitals/clinics from home were a key factor promoting self-medication [8]. Present study also highlighted that economic and financial factors promote self-medication too. Often patients are unable to meet the fee requirements of visiting doctors for which they tend to practice self-medication. A multi-

country study conducted on the economy, growth, and sustainability of the mountain area in south Asia reported that farming (39%) and agricultural work (16%) are the major occupations of most of the indigenous people living in Chittagong hill tracts [53]. The current study found most of the respondents were grouped within a monthly family income range of fewer than 236 USD (55.8%) and 236–591 USD (38.6%). A previous study in Chittagong city showed that 47% of the study participants took self-prescribed medication due to the immoderate fees of doctors, while 42% of the participants were simply reluctant to visit the doctors' chambers [23]. Our study focused on the residents of hilly areas, where it was difficult for locals to access health services. Furthermore, since health services are relatively away from residence and transport costs within these hilly regions are comparatively high, it was difficult to access health services among indigenous people who mostly rely on agriculture, cattle farming, and day labour [47, 54]. Considering all these factors, indigenous people might prefer to self-medicate unless the medical condition was serious [47, 55]. Several studies suggested that people often neglect their diseases and prefer to use self-prescribed medication instead of visiting a doctor and lack of knowledge and illiteracy influence their ignorance about health issues [56].

Antipyretics were the most used drugs as cough, cold, and fever were the most frequently occurring diseases among target populations and 53.6% of cases were self-prescribed in this study. The most self-prescribed medications were antipyretics (56.3%), analgesics (45.5%), antibiotics (34.5%), antacids and anti-ulcerants (23,4%) and antidiarrheals (17.3%). These findings are in contrast with the study conducted in in Chittagong city where, the most self-prescribed drug was antacid (38%) followed by antipyretics (21%) and antibiotics (15%) [23]. A previous study conducted among the tribal population in north Maharashtra reported similar result, that antipyretics (91.8%) and analgesics (85.7%) were the most common self-prescribed medications [27]. Also, studies in Serbia [41], Brazil [57], Pakistan [56] and coastal South India [14], and south India [40] reported analgesics and antipyretics as the most self-prescribed medications.

Antibiotics were used by 288 (21.3%) people and most of the cases (80.9%) were self-prescribed or taken without prescription. The high rate of self-medication with antibiotics among the study population exposed a great public health issue with potential side effects and being resistance to the body [58]. The study result was higher than most of the studies conducted on antibiotics self-medication such as India [14], Nigeria [59], Ghana [60], Greece [61], Tanzania [62], Jordan [63, 64], Sudan [65], Lithuania [66]. Moreover, a systemic review in the WHO south Asia region reported that most of the findings from the study conducted in these areas in different countries (Bhutan, Bangladesh, Indonesia, India, Srilanka, Nepal, Thailand, Korea) were lower than this study finding except one study among medical students in India which found a prevalence of 85.6% [67]. A large group of the population is consuming antibiotics without medical supervision which can result serious health consequences. Further study should be conducted to assess the magnitude of antibiotic misuse in the CHT area.

This study comprehends self-medication and its associated factors among the indigenous population in CHT. Misuse of antibiotics is common and poor health care knowledge and ignorance among them can lead to antibiotic resistance. This population lacks proper knowledge about the adverse effects of self-medication due to their cultural, socio-economical, geographical, and traditional distinction over the plain land population. These unprivileged people need extra care to cope with the mainstream populations. This study's findings have important policy implications. We found that self-medication was high among the indigenous population residing in the hilly regions of CHTs. The government, notably the Ministry of Chittagong Hill Tracts Affairs and the Ministry of Health and Family Welfare, should implement awareness raising initiatives among this people about the dangers of self-medication. It can be useful to arrange a campaign to prevent self-medication in collaboration with

community leaders and community health workers. Relevant authorities should also keep a close eye to improve the access to health services among this vulnerable population group. Students are the future leaders and policymakers and public health practitioners, therefore, should take appropriate efforts to improve student knowledge about self-medication with the assistance of school, college, and university administration. Use of audio and video materials can aid in this effort. More studies should be conducted to draw the attention of Government and health care organizations to mitigate issues that encourage self-medication in the less favored and marginalized populations in Bangladesh.

## Conclusion

Self-medication is a mutual problem shared among populations all around the globe. Leftover medicines, old prescriptions and sources found on the internet have been driving people to indulge into self-medication for some time. As self-medication can cause severe health complications in the long run, it is essential to take nationwide steps and measures to prevent it. Some quick steps can be enacted by chemists, where they should refrain themselves from playing doctors and profiting out of gullible individuals. Both government and non-government agencies should ensure that people living in the hill tracts receive equal medical facilities as those living in the urban areas. Campaigns should be launched to help economically poor individuals access medical help. Enactment of laws against unrestrained selling of prescription drugs is also necessary.

## Supporting information

**S1 Table. Frequencies and chi-square test for socio-demographic variables and self-medication and types of self-medication practice.**
(DOCX)

**S1 Fig. Prevalence of different types of diseases among indigenous population in CHT.** (A) Common disease prevalence among respondents; here Y-axis and the X-axis represent disease vs population percentage. (B) Common disease prevalence among districts, (C) disease prevalence among different age (years) groups and (D) disease prevalence among different BMI groups (<18.5 = underweight, 18.5–23.5 = normal weight, >23.5 = overweight); here Y-axis represents the percentage of the population and the X-axis represents districts, the age (years) of the respondents, and their BMI respectively.
(TIF)

**S2 Fig. Prevalence of common comorbidities among indigenous populations in CHT.** (A) prevalence of different types of comorbidities (B) prevalence of comorbidities based on districts (C) prevalence of comorbidities among different age (years) groups compared and (D) prevalence of comorbidities among different BMI groups (<18.5 = underweight, 18.5– 23.5 = normal weight, >23.5 = overweight); here Y-axis indicates the percentage of population and X-axis indicates types of comorbidities, living districts, respondents age (years), and their BMI respectively.
(TIF)

**S1 Dataset.**
(SAV)

**S1 File.**
(PDF)

**S2 File.**
(PDF)

## Acknowledgments

This paper is the outcome of one of the projects from the CURHS autumn internship 2020 program. The authors would like to appreciate Chittagong University Research and Higher Study Society (CURHS) for their help and support.

## Author Contributions

**Conceptualization:** Ayan Saha, Kay Kay Shain Marma.

**Data curation:** Kay Kay Shain Marma, Afrah Rashid, Srabanty Das, Prashanta Chakraborty, Sabuj Kanti Mistry.

**Formal analysis:** Ayan Saha, Nowshin Tarannum.

**Funding acquisition:** Ayan Saha.

**Investigation:** Ayan Saha, Kay Kay Shain Marma, Nowshin Tarannum.

**Methodology:** Ayan Saha, Afrah Rashid, Srabanty Das, Sabuj Kanti Mistry.

**Project administration:** Ayan Saha, H. M. Hamidullah Mehedi.

**Resources:** Ayan Saha, Srabanty Das, Tonmoy Chowdhury, Nusrat Afrin, Prashanta Chakraborty, Md. Emran, Mohammad Imdad Hussain, Ashim Barua.

**Software:** H. M. Hamidullah Mehedi, Sabuj Kanti Mistry.

**Supervision:** Ayan Saha, Sabuj Kanti Mistry.

**Validation:** Ayan Saha.

**Visualization:** Afrah Rashid.

**Writing – original draft:** Kay Kay Shain Marma, Afrah Rashid, Nowshin Tarannum.

**Writing – review & editing:** Ayan Saha, H. M. Hamidullah Mehedi, Sabuj Kanti Mistry.

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
