## [Decision Letter · Decision Letter 0]

30 Dec 2021

PONE-D-21-32179Risk factors associated with self-medication among the indigenous communities of Chittagong Hill Tracts, BangladeshPLOS ONE

Dear Dr. Saha,

Thank you for submitting your manuscript to PLOS ONE. After careful consideration, we feel that it has merit but does not fully meet PLOS ONE’s publication criteria as it currently stands. Therefore, we invite you to submit a revised version of the manuscript that addresses the points raised during the review process.

We look forward to receiving your revised manuscript.

Kind regards,

Vijayaprakash Suppiah, PhD

Academic Editor

PLOS ONE

a) Did participants provide their written or verbal informed consent to participate in this study?

4. Please remove your figures and supplementary figures from within your manuscript file, leaving only the individual TIFF/EPS image files, uploaded separately.  These will be automatically included in the reviewers’ PDF.

Reviewers' comments:

Reviewer's Responses to Questions

**Comments to the Author**

1. Is the manuscript technically sound, and do the data support the conclusions?

Reviewer #1: Partly

Reviewer #2: Yes

Reviewer #3: Yes

2. Has the statistical analysis been performed appropriately and rigorously? 

Reviewer #1: No

Reviewer #2: Yes

Reviewer #3: Yes

3. Have the authors made all data underlying the findings in their manuscript fully available?

Reviewer #1: Yes

Reviewer #2: Yes

Reviewer #3: Yes

4. Is the manuscript presented in an intelligible fashion and written in standard English?

Reviewer #1: No

Reviewer #2: Yes

Reviewer #3: Yes

5. Review Comments to the Author

Reviewer #1: The study was conducted partly on PLOS one requirement. But my opinion is that it is a duplication of work done by other researchers. Nothing innovative in the methodology. But it can be approved by considering the fact that such studies are not done in the particular population

Reviewer #2: After thoroughly going through the research article, I present my comments below.

1) The authors in the results section, represented their finding in the form of overall Odds ratio and also Adjusted Odds ratio but did not mention the factors/aspects that were adjusted for.

2) The authors in the study design section mentioned that they received insufficient responses from the participants. It would have been clearer for the readers if information regarding the insufficient response is mentioned and how the authors managed the data regarding this insufficient response.

3) Self medication practices followed by the sample population in this study includes both over the counter (OTC) medications and Prescription drugs. Combining both OTC medications and Prescription drugs to draw a conclusive outcome may be a concern for validity of the overall outcome of the study.

4) It would have been a great help if the questionnaire used in the research was sent along with the manuscript. Lack of access to the questionnaire made it difficult to understand the factors that were considered in the analysis did make the process of reviewing the research article a bit difficult.

Reviewer #3: The authors have represented the manuscript technically with satisfactory statistical analysis. Authors need to justify and elaborate the experimental findings in the discussion section more clearly. The impact (positive or negative) of education or literacy rate, economical and social standards of living on the self medication need to be discussed as the results have shown some odd findings (that need explanation) such as increase in the incidences of self medication in more educated and higher income people. In what type of problem (like social, economical, problems related to education, ethics, laws, etc.), the self medication should be included. What type of measures should be taken by the government and/ or health department? Any specific or even general preventive measures of self medication by the students as this group form the future of a country?

6. PLOS authors have the option to publish the peer review history of their article (what does this mean?). If published, this will include your full peer review and any attached files.

Reviewer #1: No

Reviewer #2: No

Reviewer #3: No

---

## [Author Response · Author response to Decision Letter 0]

29 Jan 2022

1. Thank you. The title and body of the manuscript have been modified in accordance with the PLOSONE guidelines.

2. a) Before participating in the study, participants provided verbal consent (Supporting Information file 1 and 2). It is mentioned in the section on Data Collection (Line 164).

b) i) During the data collection period, the Covid19 situation in Bangladesh was severe. We wanted to reduce any physical contact between participants and interviewers in order to ensure their safety. As a result, just verbal consent was obtained while maintaining a safe distance. ii) All the data were first manually recorded in paper (Supporting Information file 1 and 2) and then transferred to Google forms software to be exported and stored in Microsoft Excel 2013.

3. We have uploaded the data set (Supporting Information file 3) with revised manuscript for this study.

We don’t have any ethical or legal restrictions to sharing this data publicly. 

4. The manuscript's figures and supplementary figures have been removed and uploaded separately as TIFF image files.

All of the references have been double-checked. 

The following references have been added to the revised copy: 33, 49, 51, 56, and 57.

Response to Reviewer 1: 

Thank you for your comment. We acknowledge your concept that this type study was done in Bangladesh but to the best of our knowledge this is the first study that has been carried out among the indigenous population in Bangladesh. So, in this context it adds novelty to the existing literature.

Response to Reviewer 2: 

1. Thank you for bringing this to our attention. In the Statistical Analysis section (Line 173), we have gone through the details of calculating the adjusted and odd ratios.

2. Thank you for bringing this to our knowledge. We clarified the participants response number in our manuscript and rewrite the sentence in line 120.

3. We defined the self-medication as use of over-the-counter medications as well as the use of medications previously prescribed by doctor for a disease and taking the same medications for the current episode of that disease without consulting a doctor. We added this in the line 129 of the manuscript.

4. We uploaded separate file of the questionnaire (Supporting Information file 1 and 2) as supporting file. 

Response to Reviewer 3:

Thank you very much for sharing your ideas. In the revised manuscript, we have discussed how higher education and low income may influence self-medication practise among the indigenous population of the Chittagong hill region, as per your recommendation. We've cited references to back up our statements (Line 311, line 334). Our recommendations to policymakers are also mentioned in the discussion's final paragraph (Line 371).

---

## [Decision Letter · Decision Letter 1]

4 Apr 2022

PONE-D-21-32179R1Risk factors associated with self-medication among the indigenous communities of Chittagong Hill Tracts, BangladeshPLOS ONE

Dear Dr. Saha,

Thank you for submitting your manuscript to PLOS ONE. After careful consideration, we feel that it has merit but does not fully meet PLOS ONE’s publication criteria as it currently stands. Therefore, we invite you to submit a revised version of the manuscript that addresses the points raised during the review process.

While the reviewers felt that your manuscript has largely improved, Reviewer 2 felt that further clarification is needed regarding the calculation of odds ratio and adjusted odds ratio. As such, we ask you to revise the manuscript to address the reviewer's specific comments.

We look forward to receiving your revised manuscript.

Kind regards,

Natasha McDonald, PhD

Associate Editor

PLOS ONE

Journal Requirements:

Reviewers' comments:

Reviewer's Responses to Questions

**Comments to the Author**

1. If the authors have adequately addressed your comments raised in a previous round of review and you feel that this manuscript is now acceptable for publication, you may indicate that here to bypass the “Comments to the Author” section, enter your conflict of interest statement in the “Confidential to Editor” section, and submit your "Accept" recommendation.

Reviewer #1: All comments have been addressed

Reviewer #2: All comments have been addressed

Reviewer #3: (No Response)

2. Is the manuscript technically sound, and do the data support the conclusions?

Reviewer #1: Yes

Reviewer #2: Partly

Reviewer #3: Yes

3. Has the statistical analysis been performed appropriately and rigorously? 

Reviewer #1: Yes

Reviewer #2: Yes

Reviewer #3: Yes

4. Have the authors made all data underlying the findings in their manuscript fully available?

Reviewer #1: Yes

Reviewer #2: Yes

Reviewer #3: Yes

5. Is the manuscript presented in an intelligible fashion and written in standard English?

Reviewer #1: Yes

Reviewer #2: Yes

Reviewer #3: Yes

6. Review Comments to the Author

Reviewer #1: (No Response)

Reviewer #2: With respect to the response given by the authors to the first comment, they mentioned the statistical method used for calculation of odds ratio and adjusted odds ratio. But what is actually asked for are the factors that were considered while adjusting the odds ratio. For example, what factors were considered for adjusted odds ratio of 0.60 in 18-35 year age group when compared to the crude odds ratio of 1.92. (Table 3)

Reviewer #3: (No Response)

7. PLOS authors have the option to publish the peer review history of their article (what does this mean?). If published, this will include your full peer review and any attached files.

Reviewer #1: **Yes: **ANTRIYA ANNIE TOM

Reviewer #2: No

Reviewer #3: No

---

## [Author Response · Author response to Decision Letter 1]

14 Apr 2022

To reviewer two:

Thanks for your comment. For your clarification the model building approach is presented in Methods section as follows:

“The variables with a p value of less than 0.25 in the unadjusted analysis was included in the final multiple regression model.” (Please see line 173-174).

We have also clarified this issue in more detail in the Result section as follows:

“All the variables had a p value less than 0.25 in the bivariate analysis and thus all of them were included in the multivariable model. Hence, the model is adjusted for all the variables in Table 3.” (Please see line 270-272).

---

## [Decision Letter · Decision Letter 2]

13 May 2022

PONE-D-21-32179R2Risk factors associated with self-medication among the indigenous communities of Chittagong Hill Tracts, BangladeshPLOS ONE

Dear Dr. Saha,

Thank you for submitting your manuscript to PLOS ONE. After careful consideration, we feel that it has merit but does not fully meet PLOS ONE’s publication criteria as it currently stands. Therefore, we invite you to submit a revised version of the manuscript that addresses the points raised during the review process.

The reviewers are generally satisfied with the revisions you have made to your manuscript; however, one reviewer raised the need for clarification of a minor point in the statistical reporting. Please see their comment below and amend your manuscript to clarify this point.

We look forward to receiving your revised manuscript.

Kind regards,

Natasha McDonald, PhD

Associate Editor

PLOS ONE

Journal Requirements:

Reviewers' comments:

Reviewer's Responses to Questions

**Comments to the Author**

1. If the authors have adequately addressed your comments raised in a previous round of review and you feel that this manuscript is now acceptable for publication, you may indicate that here to bypass the “Comments to the Author” section, enter your conflict of interest statement in the “Confidential to Editor” section, and submit your "Accept" recommendation.

Reviewer #2: All comments have been addressed

2. Is the manuscript technically sound, and do the data support the conclusions?

Reviewer #2: Partly

3. Has the statistical analysis been performed appropriately and rigorously? 

Reviewer #2: Yes

4. Have the authors made all data underlying the findings in their manuscript fully available?

Reviewer #2: Yes

5. Is the manuscript presented in an intelligible fashion and written in standard English?

Reviewer #2: Yes

6. Review Comments to the Author

Reviewer #2: Thank you for the response for the comments provided. Now all the comments are addressed and explained. The authors mentioned that the variables whose p-value is less than 0.25 were considered in the final analysis of adjusted odds ratios. So all the parameters mentioned in table.2 were considered while deriving the adjusted odd's ratio mentioned in table 3?

7. PLOS authors have the option to publish the peer review history of their article (what does this mean?). If published, this will include your full peer review and any attached files.

Reviewer #2: No

---

## [Author Response · Author response to Decision Letter 2]

14 May 2022

Yes, we have considered all the socio-demographic characteristics of the participants presented in table 1. We performed bivariate analysis with each of them and those of p value less than 0.25 in the bivariate analysis were included in the multiple regression model. We have clarified this in the revised manuscript. Please see page 21 line 261.

---

## [Editor Report · Decision Letter 3]

25 May 2022

Risk factors associated with self-medication among the indigenous communities of Chittagong Hill Tracts, Bangladesh

PONE-D-21-32179R3

Dear Dr. Saha,

We’re pleased to inform you that your manuscript has been judged scientifically suitable for publication and will be formally accepted for publication once it meets all outstanding technical requirements.

Kind regards,

Carla Pegoraro

Division Editor

PLOS ONE

Additional Editor Comments (optional):

Thank you for addressing the last minor point and clarifying your submission.

---

## [Editor Report · Acceptance letter]

3 Jun 2022

PONE-D-21-32179R3 

Risk factors associated with self-medication among the indigenous communities of Chittagong Hill Tracts, Bangladesh 

Dear Dr. Saha:

I'm pleased to inform you that your manuscript has been deemed suitable for publication in PLOS ONE. Congratulations! Your manuscript is now with our production department. 

Kind regards, 

on behalf of

Dr Carla Pegoraro 

Staff Editor

PLOS ONE